# Max-Margin Majority Voting for Learning from Crowds

**Tian Tian,  Jun Zhu**

Department of Computer Science & Technology; Center for Bio-Inspired Computing Research
Tsinghua National Lab for Information Science & Technology
State Key Lab of Intelligent Technology & Systems; Tsinghua University, Beijing 100084, China
`tiant13@mails.tsinghua.edu.cn; dcszj@tsinghua.edu.cn`

## Abstract

Learning-from-crowds aims to design proper aggregation strategies to infer the unknown true labels from the noisy labels provided by ordinary web workers. This paper presents max-margin majority voting ($M^3V$) to improve the discriminative ability of majority voting and further presents a Bayesian generalization to incorporate the flexibility of generative methods on modeling noisy observations with worker confusion matrices. We formulate the joint learning as a regularized Bayesian inference problem, where the posterior regularization is derived by maximizing the margin between the aggregated score of a potential true label and that of any alternative label. Our Bayesian model naturally covers the Dawid-Skene estimator and $M^3V$. Empirical results demonstrate that our methods are competitive, often achieving better results than state-of-the-art estimators.

## 1   Introduction

Many learning tasks require labeling large datasets. Though reliable, it is often too expensive and time-consuming to collect labels from domain experts or well-trained workers. Recently, online crowdsourcing platforms have dramatically decreased the labeling cost by dividing the workload into small parts, then distributing micro-tasks to a crowd of ordinary web workers [17, 20]. However, the labeling accuracy of web workers could be lower than expected due to their various backgrounds or lack of knowledge. To improve the accuracy, it is usually suggested to label every task multiple times by different workers, then the redundant labels can provide hints on resolving the true labels.

Much progress has been made in designing effective aggregation mechanisms to infer the true labels from noisy observations. From a modeling perspective, existing work includes both generative approaches and discriminative approaches. A generative method builds a flexible probabilistic model for generating the noisy observations conditioned on the unknown true labels and some behavior assumptions, with examples of the Dawid-Skene (DS) estimator [5], the minimax entropy (Entropy) estimator[1] [24, 25], and their variants. In contrast, a discriminative approach does not model the observations; it directly identifies the true labels via some aggregation rules. Examples include majority voting and the weighted majority voting that takes worker reliability into consideration [10, 11].

In this paper, we present a max-margin formulation of the most popular majority voting estimator to improve its discriminative ability, and further present a Bayesian generalization that conjoins the advantages of both generative and discriminative approaches. The max-margin majority voting ($M^3V$) directly maximizes the margin between the aggregated score of a potential true label and that of any alternative label, and the Bayesian model consists of a flexible probabilistic model to generate the noisy observations by conditioning on the unknown true labels. We adopt the same approach as the

classical Dawid-Skene estimator to build the probabilistic model by considering worker confusion matrices, though many other generative models are also possible. Then, we strongly couple the generative model and $M^3V$ by formulating a joint learning problem under the regularized Bayesian inference (RegBayes) [27] framework, where the posterior regularization [7] enforces a large margin between the potential true label and any alternative label. Naturally, our Bayesian model covers both the David-Skene estimator and $M^3V$ as special cases by setting the regularization parameter to its extreme values (i.e., 0 or $\infty$). We investigate two choices on defining the max-margin posterior regularization: (1) an *averaging* model with a variational inference algorithm; and (2) a *Gibbs* model with a Gibbs sampler under a data augmentation formulation. The averaging version can be seen as an extension to the MLE learner of Dawid-Skene model. Experiments on real datasets suggest that max-margin learning can significantly improve the accuracy of majority voting, and that our Bayesian estimators are competitive, often achieving better results than state-of-the-art estimators on true label estimation tasks.

## 2  Preliminary

We consider the label aggregation problem with a dataset consisting of $M$ items (e.g., pictures or paragraphs). Each item $i$ has an unknown true label $y_i \in [D]$, where $[D] := \{1, \ldots, D\}$. The task $t_i$ is to label item $i$. In crowdsourcing, we have $N$ workers assigning labels to these items. Each worker may only label a part of the dataset. Let $\mathcal{I}_i \subseteq [N]$ denote the workers who have done task $t_i$. We use $x_{ij}$ to denote the label of $t_i$ provided by worker $j$, $\boldsymbol{x}_i$ to denote the labels provided to task $t_i$, and $\boldsymbol{X}$ is the collection of these worker labels, which is an incomplete matrix. The goal of learning-from-crowds is to estimate the true labels of items from the noisy observations $\boldsymbol{X}$.

### 2.1  Majority Voting Estimator

Majority voting (MV) is arguably the simplest method. It posits that for every task the true label is always most commonly given. Thus, it selects the most frequent label for each task as its true label, by solving the problem:

$$\hat{y}_i = \operatorname*{argmax}_{d \in [D]} \sum_{j=1}^{N} \mathbb{I}(x_{ij} = d), \forall i \in [M], \tag{1}$$

where $\mathbb{I}(\cdot)$ is an indicator function. It equals to 1 whenever the predicate is true, otherwise it equals to 0. Previous work has extended this method to weighted majority voting (WMV) by putting different weights on workers to measure worker reliability [10, 11].

### 2.2  Dawid-Skene Estimator

The method of Dawid and Skene [5] is a generative approach by considering worker confusability. It posits that the performance of a worker is consistent across different tasks, as measured by a confusion matrix whose diagonal entries denote the probability of assigning correct labels while off-diagonal entries denote the probability of making specific mistakes to label items in one category as another. Formally, let $\boldsymbol{\phi}_j$ be the confusion matrix of worker $j$. Then, $\phi_{jkd}$ denotes the probability that worker $j$ assigns label $d$ to an item whose true label is $k$. Under the basic assumption that workers finish each task independently, the likelihood of observed labels can be expressed as

$$p(\boldsymbol{X}|\boldsymbol{\Phi}, \boldsymbol{y}) = \prod_{i=1}^{M} \prod_{j=1}^{N} \prod_{d,k=1}^{D} \phi_{jkd}^{n_{jkd}^{i}} = \prod_{j=1}^{N} \prod_{d,k=1}^{D} \phi_{jkd}^{n_{jkd}}, \tag{2}$$

where $n_{jkd}^{i} = \mathbb{I}(x_{ij} = d, y_i = k)$, and $n_{jkd} = \sum_{i=1}^{M} n_{jkd}^{i}$ is the number of tasks with true label $k$ but being labeled to $d$ by worker $j$.

The unknown labels and parameters can be estimated by maximum-likelihood estimation (MLE), $\{\hat{\boldsymbol{y}}, \hat{\boldsymbol{\Phi}}\} = \operatorname{argmax}_{\boldsymbol{y}, \boldsymbol{\Phi}} \log p(\boldsymbol{X}|\boldsymbol{\Phi}, \boldsymbol{y})$, via an expectation-maximization (EM) algorithm that iteratively updates the true labels $\boldsymbol{y}$ and the parameters $\boldsymbol{\Phi}$. The learning procedure is often initialized by majority voting to avoid bad local optima. If we assume some structure of the confusion matrix, various variants of the DS estimator have been studied, including the homogenous DS model [15] and the class-conditional DS model [11]. We can also put a prior over worker confusion matrices and transform the inference into a standard inference problem in graphical models [12]. Recently, spectral methods have also been applied to better initialize the DS model [23].

## 3 Max-Margin Majority Voting

Majority voting is a discriminative model that directly finds the most likely label for each item. In this section, we present max-margin majority voting ($M^3V$), a novel extension of (weighted) majority voting with a new notion of margin (named crowdsourcing margin).

### 3.1 Geometric Interpretation of Crowdsourcing Margin

Let $g(x_i, d)$ be a $N$-dimensional vector, with the element $j$ equaling to $\mathbb{I}(j \in \mathcal{I}_i, x_{ij} = d)$. Then, the estimation of the vanilla majority voting in Eq. (1) can be formulated as finding solutions $\{y_i\}_{i \in [M]}$ that satisfy the following constraints:

$$\mathbf{1}_N^\top g(x_i, y_i) - \mathbf{1}_N^\top g(x_i, d) \geq 0, \quad \forall i, d, \quad (3)$$

where $\mathbf{1}_N$ is the $N$-dimensional all-one vector and $\mathbf{1}_N^\top g(x_i, k)$ is the aggregated score of the potential true label $k$ for task $t_i$. By using the all-one vector, the aggregated score has an intuitive interpretation — it denotes the number of workers who have labeled $t_i$ as class $k$.

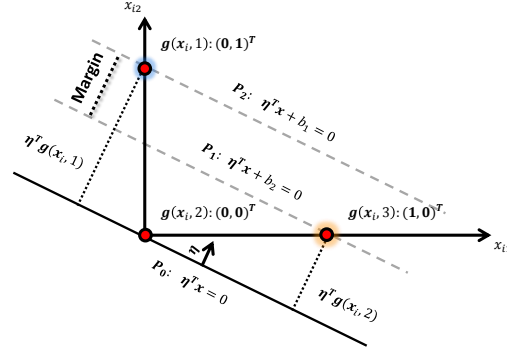

Figure 1: A geometric interpretation of the crowdsourcing margin.

Apparently, the all-one vector treats all workers equally, which may be unrealistic in practice due to the various backgrounds of the workers. By simply choosing what the majority of workers agree on, the vanilla MV is prone to errors when many workers give low quality labels. One way to tackle this problem is to take worker reliability into consideration. Let $\eta$ denote the worker weights. When these values are known, we can get the aggregated score $\eta^\top g(x_i, k)$ of a weighted majority voting (WMV), and estimate the true labels by the rule: $\hat{y}_i = \arg\max_{d \in [D]} \eta^\top g(x_i, d)$. Thus, reliable workers contribute more to the decisions.

Geometrically, $g(x_i, d)$ is a point in the $N$-dimensional space for each task $t_i$. The aggregated score $\mathbf{1}_N^\top g(x_i, d)$ measures the distance (up to a constant scaling) from this point to the hyperplane $\mathbf{1}_N^\top x = 0$. So the MV estimator actually finds a point that has the largest distance to that hyperplane for each task, and the decision boundary of majority voting is another hyperplane $\mathbf{1}_N^\top x - b = 0$ which separates the point $g(x_i, \hat{y}_i)$ from the other points $g(x_i, k)$, $k \neq \hat{y}_i$. By introducing the worker weights $\eta$, we relax the constraint of the all-one vector to allow for more flexible decision boundaries $\eta^\top x - b = 0$. All the possible decision boundaries with the same orientation are equivalent. Inspired by the generalized notion of margin in multi-class SVM [4], we define the *crowdsourcing margin* as the minimal difference between the aggregated score of the potential true label and the aggregated scores of other alternative labels. Then, one reasonable choice of the best hyperplane (i.e. $\eta$) is the one that represents the largest margin between the potential true label and other alternatives.

Fig. 1 provides an illustration of the crowdsourcing margin for WMV with $D = 3$ and $N = 2$, where each axis represents the label of a worker. Assume that both workers provide labels 3 and 1 to item $i$. Then, the vectors $g(x_i, y)$, $y \in [3]$ are three points in the 2D plane. Given the worker weights $\eta$, the estimated label should be 1, since $g(x_i, 1)$ has the largest distance to line $P_0$. Line $P_1$ and line $P_2$ are two boundaries that separate $g(x_i, 1)$ and other points. The margin is the distance between them. In this case, $g(x_i, 1)$ and $g(x_i, 3)$ are support vectors that decide the margin.

### 3.2 Max-Margin Majority Voting Estimator

Let $\ell$ be the minimum margin between the potential true label and all other alternatives. We define the *max-margin majority voting* ($M^3V$) as solving the constrained optimization problem to estimate the true labels $y$ and weights $\eta$:

$$\inf_{\eta, y} \quad \frac{1}{2} \|\eta\|_2^2 \qquad (4)$$

$$\text{s.t.}: \eta^\top g_i^\Delta(d) \geq \ell_i^\Delta(d), \forall i \in [M], d \in [D],$$

where $g_i^\Delta(d) := g(x_i, y_i) - g(x_i, d)$ [2] and $\ell_i^\Delta(d) = \ell \mathbb{I}(y_i \neq d)$. And in practice, the worker labels are often linearly inseparable by a single hyperplane. Therefore, we relax the hard constraints

by introducing non-negative slack variables $\{\xi_i\}_{i=1}^M$, one for each task, and define the soft-margin max-margin majority voting as

$$\inf_{\xi_i \geq 0, \boldsymbol{\eta}, \boldsymbol{y}} \quad \frac{1}{2}\|\boldsymbol{\eta}\|_2^2 + c\sum_i \xi_i \tag{5}$$

$$\text{s.\,t.}: \boldsymbol{\eta}^\top \boldsymbol{g}_i^\Delta(d) \geq \ell_i^\Delta(d) - \xi_i, \forall i \in [M], d \in [D],$$

where $c$ is a positive regularization parameter and $\ell - \xi_i$ is the soft-margin for task $t_i$. The value of $\xi_i$ reflects the difficulty of task $t_i$ — a small $\xi_i$ suggests a large discriminant margin, indicating that the task is easy with a rare chance to make mistakes; while a large $\xi_i$ suggests that the task is hard with a higher chance to make mistakes. Note that our max-margin majority voting is significantly different from the unsupervised SVMs (or max-margin clustering) [21], which aims to assign cluster labels to the data points by maximizing some different notion of margin with balance constraints to avoid trivial solutions. Our M³V does not need such balance constraints.

Albeit not jointly convex, problem (5) can be solved by iteratively updating $\boldsymbol{\eta}$ and $\boldsymbol{y}$ to find a local optimum. For $\boldsymbol{\eta}$, the solution can be derived as $\boldsymbol{\eta} = \sum_{i=1}^M \sum_{d=1}^D \omega_i^d \boldsymbol{g}_i^\Delta(d)$ by the fact that the subproblem is convex. The parameters $\boldsymbol{\omega}$ are obtained by solving the dual problem

$$\sup_{0 \leq \omega_i^d \leq c} -\frac{1}{2}\boldsymbol{\eta}^\top \boldsymbol{\eta} + \sum_i \sum_d \omega_i^d \ell_i^\Delta(d), \tag{6}$$

which is exactly the QP dual problem in standard SVM [4]. So it can be efficiently solved by well-developed SVM solvers like LIBSVM [2]. For updating $\boldsymbol{y}$, we define $(x)_+ := \max(0, x)$, and then it is a weighted majority voting with a margin gap constraint:

$$\hat{y}_i = \underset{y_i \in [D]}{\operatorname{argmax}} \left( -c \max_{d \in [D]} \left( \ell_i^\Delta(d) - \hat{\boldsymbol{\eta}}^\top \boldsymbol{g}_i^\Delta(d) \right)_+ \right), \tag{7}$$

Overall, the algorithm is a max-margin iterative weighted majority voting (MM-IWMV). Comparing with the iterative weighted majority voting (IWMV) [11], which tends to maximize the expected gap of the aggregated scores under the Homogenous DS model, our M³V directly maximizes the data specified margin without further assumption on data model. Empirically, as we shall see, our M³V could have more powerful discriminative ability with better accuracy than IWMV.

## 4 Bayesian Max-Margin Estimator

With the intuitive and simple max-margin principle, we now present a more sophisticated Bayesian max-margin estimator, which conjoins the discriminative ability of M³V and the flexibility of the generative DS estimator. Though slightly more complicated in learning and inference, the Bayesian models retain the intuitive simplicity of M³V and the flexibility of DS, as explained below.

### 4.1 Model Definition

We adopt the same DS model to generate observations conditioned on confusion matrices, with the full likelihood in Eq. (2). We further impose a prior $p_0(\boldsymbol{\Phi}, \boldsymbol{\eta})$ for Bayesian inference. Assuming that the true labels $\boldsymbol{y}$ are given, we aim to get the target posterior $p(\boldsymbol{\Phi}, \boldsymbol{\eta}|\boldsymbol{X}, \boldsymbol{y})$, which can be obtained by solving an optimization problem:

$$\inf_{q(\boldsymbol{\Phi}, \boldsymbol{\eta})} \quad \mathcal{L}\left(q(\boldsymbol{\Phi}, \boldsymbol{\eta}); \boldsymbol{y}\right), \tag{8}$$

where $\mathcal{L}(q; \boldsymbol{y}) := \mathrm{KL}(q\|p_0(\boldsymbol{\Phi}, \boldsymbol{\eta})) - \mathbb{E}_q[\log p(\boldsymbol{X}|\boldsymbol{\Phi}, \boldsymbol{y})]$ measures the Kullback-Leibler (KL) divergence between a desired post-data posterior $q$ and the original Bayesian posterior, and $p_0(\boldsymbol{\Phi}, \boldsymbol{\eta})$ is the prior, often factorized as $p_0(\boldsymbol{\Phi})p_0(\boldsymbol{\eta})$. As we shall see, this Bayesian DS estimator often leads to better performance than the vanilla DS.

Then, we explore the ideas of regularized Bayesian inference (RegBayes) [27] to incorporate max-margin majority voting constraints as posterior regularization on problem (8), and define the Bayesian max-margin estimator (denoted by CrowdSVM) as solving:

$$\inf_{\xi_i \geq 0, q \in \mathcal{P}, \boldsymbol{y}} \quad \mathcal{L}(q(\boldsymbol{\Phi}, \boldsymbol{\eta}); \boldsymbol{y}) + c \cdot \sum_i \xi_i \tag{9}$$

$$\text{s.\,t.}: \mathbb{E}_q[\boldsymbol{\eta}^\top \boldsymbol{g}_i^\Delta(d)] \geq \ell_i^\Delta(d) - \xi_i, \forall i \in [M], d \in [D],$$

where $\mathcal{P}$ is the probabilistic simplex, and we take expectation over $q$ to define the margin constraints. Such posterior constraints will influence the estimates of $\boldsymbol{y}$ and $\boldsymbol{\Phi}$ to get better aggregation, as we shall see. We use a Dirichlet prior on worker confusion matrices, $\boldsymbol{\phi}_{mk}|\boldsymbol{\alpha} \sim \mathrm{Dir}(\boldsymbol{\alpha})$, and a spherical Gaussian prior on $\boldsymbol{\eta}$, $\boldsymbol{\eta} \sim \mathcal{N}(\mathbf{0}, v\boldsymbol{I})$. By absorbing the slack variables, CrowdSVM solves the equivalent unconstrained problem:

$$\inf_{q \in \mathcal{P}, \boldsymbol{y}} \mathcal{L}(q(\boldsymbol{\Phi}, \boldsymbol{\eta}); \boldsymbol{y}) + c \cdot \mathcal{R}_m(q(\boldsymbol{\Phi}, \boldsymbol{\eta}); \boldsymbol{y}), \tag{10}$$

where $\mathcal{R}_m(q; \boldsymbol{y}) = \sum_{i=1}^{M} \max_{d=1}^{D} \left( \ell_i^{\Delta}(d) - \mathbb{E}_q[\boldsymbol{\eta}^{\top} \boldsymbol{g}_i^{\Delta}(d)] \right)_+$ is the posterior regularization.

**Remark 1.** *From the above definition, we can see that both the Bayesian DS estimator and the max-margin majority voting are special cases of CrowdSVM. Specifically, when $c \to 0$, it is equivalent to the DS model. If we set $v = v'/c$ for some positive parameter $v'$, then when $c \to \infty$ CrowdSVM reduces to the max-margin majority voting.*

## 4.2 Variational Inference

Since it is intractable to directly solve problem (9) or (10), we introduce the structured mean-field assumption on the post-data posterior, $q(\boldsymbol{\Phi}, \boldsymbol{\eta}) = q(\boldsymbol{\Phi})q(\boldsymbol{\eta})$, and solve the problem by alternating minimization as outlined in Alg. 1. The algorithm iteratively performs the following steps until a local optimum is reached:

---
**Algorithm 1:** The CrowdSVM algorithm

1. Initialize $y$ by majority voting.
**while** *Not converge* **do**
  2. For each worker $j$ and category $k$:
    $q(\boldsymbol{\phi}_{jk}) \leftarrow \mathrm{Dir}(\boldsymbol{n}_{jk} + \boldsymbol{\alpha})$.
  3. Solve the dual problem (11).
  4. For each item $i$: $\hat{y}_i \leftarrow \mathrm{argmax}_{y_i \in [D]} f(y_i, \boldsymbol{x}_i; q)$.
**end**

---

**Infer $q(\boldsymbol{\Phi})$:** Fixing the distribution $q(\boldsymbol{\eta})$ and the true labels $\boldsymbol{y}$, the problem in Eq. (9) turns to a standard Bayesian inference problem with the closed-form solution: $q^*(\boldsymbol{\Phi}) \propto p_0(\boldsymbol{\Phi})p(\boldsymbol{X}|\boldsymbol{\Phi}, \boldsymbol{y})$. Since the prior is a Dirichlet distribution, the inferred distribution is also Dirichlet, $q^*(\boldsymbol{\phi}_{jk}) = \mathrm{Dir}(\boldsymbol{n}_{jk} + \boldsymbol{\alpha})$, where $\boldsymbol{n}_{jk}$ is a $D$-dimensional vector with element $d$ being $n_{jkd}$.

**Infer $q(\boldsymbol{\eta})$ and solve for $\boldsymbol{\omega}$:** Fixing the distribution $q(\boldsymbol{\Phi})$ and the true labels $\boldsymbol{y}$, we optimize Eq. (9) over $q(\boldsymbol{\eta})$, which is also convex. We can derive the optimal solution: $q^*(\boldsymbol{\eta}) \propto p_0(\boldsymbol{\eta}) \exp\left( \boldsymbol{\eta}^{\top} \sum_i \sum_d \omega_i^d \boldsymbol{g}_i^{\Delta}(d) \right)$, where $\boldsymbol{\omega} = \{\omega_i^d\}$ are Lagrange multipliers. With the normal prior, $p_0(\boldsymbol{\eta}) = \mathcal{N}(0, v\boldsymbol{I})$, the posterior is a normal distribution: $q^*(\boldsymbol{\eta}) = \mathcal{N}(\boldsymbol{\mu}, v\boldsymbol{I})$, whose mean is $\boldsymbol{\mu} = v \sum_{i=1}^{M} \sum_{d=1}^{D} \omega_i^d \boldsymbol{g}_i^{\Delta}(d)$. Then the parameters $\boldsymbol{\omega}$ are obtained by solving the dual problem

$$\sup_{0 \leq \omega_i^d \leq c} -\frac{1}{2v} \boldsymbol{\mu}^{\top} \boldsymbol{\mu} + \sum_i \sum_d \omega_i^d \ell_i^{\Delta}(d), \tag{11}$$

which is same as the problem (6) in max-margin majority voting.

**Infer $\boldsymbol{y}$:** Fixing the distributions of $\boldsymbol{\Phi}$ and $\boldsymbol{\eta}$ at their optimum $q^*$, we find $\boldsymbol{y}$ by solving problem (10). To make the prediction more efficient, we approximate the distribution $q^*(\boldsymbol{\Phi})$ by a Dirac delta mass $\delta(\boldsymbol{\Phi} - \hat{\boldsymbol{\Phi}})$, where $\hat{\boldsymbol{\Phi}}$ is the mean of $q^*(\boldsymbol{\Phi})$. Then since all tasks are independent, we can derive the discriminant function of $y_i$ as

$$f(y_i, \boldsymbol{x}_i; q^*) = \log p(\boldsymbol{x}_i|\hat{\boldsymbol{\Phi}}, y_i) - c \max_{d \in [D]} \left( (\ell_i^{\Delta}(d) - \hat{\boldsymbol{\mu}}^{\top} \boldsymbol{g}_i^{\Delta}(d))_+ \right), \tag{12}$$

where $\hat{\boldsymbol{\mu}}$ is the mean of $q^*(\boldsymbol{\eta})$. Then we can make predictions by maximize this function.

Apparently, the discriminant function (12) represents a strong coupling between the generative model and the discriminative margin constraints. Therefore, CrowdSVM jointly considers these two factors when estimating true labels. We also note that the estimation rule used here reduces to the rule (7) of MM-IWMV by simply setting $c = \infty$.

## 5 Gibbs CrowdSVM Estimator

CrowdSVM adopts an averaging model to define the posterior constraints in problem (9). Here, we further provide an alternative strategy which leads to a full Bayesian model with a Gibbs sampler. The resulting Gibbs-CrowdSVM does not need to make the mean-field assumption.

## 5.1 Model Definition

Suppose the target posterior $q(\mathbf{\Phi}, \boldsymbol{\eta})$ is given, we perform the max-margin majority voting by drawing a random sample $\boldsymbol{\eta}$. This leads to the crowdsourcing hinge-loss

$$\mathcal{R}(\boldsymbol{\eta}, \boldsymbol{y}) = \sum_{i=1}^{M} \max_{d \in [D]} \left( \ell_i^{\Delta}(d) - \boldsymbol{\eta}^{\top} \boldsymbol{g}_i^{\Delta}(d) \right)_+, \tag{13}$$

which is a function of $\boldsymbol{\eta}$. Since $\boldsymbol{\eta}$ are random, we define the overall hinge-loss as the expectation over $q(\boldsymbol{\eta})$, that is, $\mathcal{R'}_m(q(\mathbf{\Phi}, \boldsymbol{\eta}); \boldsymbol{y}) = \mathbb{E}_q\left[\mathcal{R}(\boldsymbol{\eta}, \boldsymbol{y})\right]$. Due to the convexity of $\max$ function, the expected loss is in fact an upper bound of the average loss, i.e., $\mathcal{R'}_m(q(\mathbf{\Phi}, \boldsymbol{\eta}); \boldsymbol{y}) \geq \mathcal{R}_m(q(\mathbf{\Phi}, \boldsymbol{\eta}); \boldsymbol{y})$. Differing from CrowdSVM, we also treat the hidden true labels $\boldsymbol{y}$ as random variables with a uniform prior. Then we define Gibbs-CrowdSVM as solving the problem:

$$\inf_{q \in \mathcal{P}} \quad \mathcal{L}\left( q(\mathbf{\Phi}, \boldsymbol{\eta}, \boldsymbol{y}) \right) + \mathbb{E}_q\left[ \sum_{i=1}^{M} 2c(\zeta_{is_i})_+ \right], \tag{14}$$

where $\zeta_{id} = \ell_i^{\Delta}(d) - \boldsymbol{\eta}^{\top} \boldsymbol{g}_i^{\Delta}(d)$, $s_i = \operatorname{argmax}_{d \neq y_i} \zeta_{id}$, and the factor 2 is introduced for simplicity.

**Data Augmentation** In order to build an efficient Gibbs sampler for this problem, we derive the posterior distribution with the data augmentation [3, 26] for the max-margin regularization term. We let $\psi(y_i|\boldsymbol{x}_i, \boldsymbol{\eta}) = \exp(-2c(\zeta_{is_i})_+)$ to represent the regularizer. According to the equality: $\psi(y_i|\boldsymbol{x}_i, \boldsymbol{\eta}) = \int_0^{\infty} \psi(y_i, \lambda_i|\boldsymbol{x}_i, \boldsymbol{\eta}) d\lambda_i$, where $\psi(y_i, \lambda_i|\boldsymbol{x}_i, \boldsymbol{\eta}) = (2\pi\lambda_i)^{-\frac{1}{2}} \exp(\frac{-1}{2\lambda_i}(\lambda_i + c\zeta_{is_i})^2)$ is a (unnormalized) joint distribution of $y_i$ and the augmented variable $\lambda_i$ [14], the posterior of Gibbs-CrowdSVM can be expressed as the marginal of a higher dimensional distribution, i.e., $q(\mathbf{\Phi}, \boldsymbol{\eta}, \boldsymbol{y}) = \int q(\mathbf{\Phi}, \boldsymbol{\eta}, \boldsymbol{y}, \boldsymbol{\lambda}) d\boldsymbol{\lambda}$, where

$$q(\mathbf{\Phi}, \boldsymbol{\eta}, \boldsymbol{y}, \boldsymbol{\lambda}) \propto p_0(\mathbf{\Phi}, \boldsymbol{\eta}, \boldsymbol{y}) \prod_{i=1}^{M} p(\boldsymbol{x}_i|\mathbf{\Phi}, y_i)\psi(y_i, \lambda_i|\boldsymbol{x}_i, \boldsymbol{\eta}). \tag{15}$$

Putting the last two terms together, we can view $q(\mathbf{\Phi}, \boldsymbol{\eta}, \boldsymbol{y}, \boldsymbol{\lambda})$ as a standard Bayesian posterior, but with the unnormalized likelihood $\widetilde{p}(\boldsymbol{x}_i, \lambda_i|\mathbf{\Phi}, \boldsymbol{\eta}, y_i) \propto p(\boldsymbol{x}_i|\mathbf{\Phi}, y_i)\psi(y_i, \lambda_i|\boldsymbol{x}_i, \boldsymbol{\eta})$, which jointly considers the noisy observations and the large margin discrimination between the potential true labels and alternatives.

## 5.2 Posterior Inference

With the augmented representation, we can do Gibbs sampling to infer the posterior distribution $q(\mathbf{\Phi}, \boldsymbol{\eta}, \boldsymbol{y}, \boldsymbol{\lambda})$ and thus $q(\mathbf{\Phi}, \boldsymbol{\eta}, \boldsymbol{y})$ by discarding $\boldsymbol{\lambda}$. The conditional distributions for $\{\mathbf{\Phi}, \boldsymbol{\eta}, \boldsymbol{\lambda}, \boldsymbol{y}\}$ are derived in Appendix A. Note that when sample $\boldsymbol{\lambda}$ from the inverse Gaussian distribution, a fast sampling algorithm [13] can be applied with $\mathcal{O}(1)$ time complexity. And for the hidden variables $\boldsymbol{y}$, we initially set them as the results of majority voting. After removing burn-in samples, we use their most frequent values of as the final outputs.

# 6 Experiments

We now present experimental results to demonstrate the strong discriminative ability of max-margin majority voting and the promise of our Bayesian models, by comparing with various strong competitors on multiple real datasets.

## 6.1 Datasets and Setups

We use four real world crowd labeling datasets as summarized in Table 1. **Web Search** [24]: 177 workers are asked to rate a set of 2,665 query-URL pairs on a relevance rating scale from 1 to 5. Each task is labeled by 6 workers on average. In total 15,567 labels are collected. **Age** [8]: It consists of 10,020 labels of age estimations for 1,002 face images. Each image was labeled by 10 workers. And there are 165 workers involved in these tasks. The final estimations are discretized into 7 bins. **Bluebirds** [19]: It consists of 108 bluebird pictures. There are 2 breeds among all the images, and each image is labeled by all 39 workers. 4,214 labels in total. **Flowers** [18]: It contains 2,366 binary labels for a dataset with 200 flower pictures. Each worker is asked to answer whether the flower in picture is peach flower. 36 workers participate in these tasks.

We compare M³V, as well as its Bayesian extensions CrowdSVM and Gibbs-CrowdSVM, with various baselines, including majority voting (MV), iterative weighted majority voting (IWMV) [11], the Dawid-Skene (DS) estimator [5], and the minimax entropy (Entropy) es-

Table 1: Datasets Overview.

| DATASET | LABELS | ITEMS | WORKERS |
|---|---|---|---|
| WEB SEARCH | 15,567 | 2,665 | 177 |
| AGE | 10,020 | 1,002 | 165 |
| BLUEBIRDS | 4,214 | 108 | 39 |
| FLOWERS | 2,366 | 200 | 36 |

timator [25]. For Entropy estimator, we use the implementation provided by the authors, and show both the performances of its multiclass version (Entropy (M)) and the ordinal version (Entropy (O)). All the estimators that require an iterative updating are initialized by majority voting to avoid bad local minima. All experiments were conducted on a PC with Intel Core i5 3.00GHz CPU and 12.00GB RAM.

## 6.2 Model Selection

Due to the special property of crowdsourcing, we cannot simply split the training data into multiple folds to cross-validate the hyperparameters by using accuracy as the selection criterion, which may bias to over-optimistic models. Instead, we adopt the likelihood $p(\boldsymbol{X}|\hat{\boldsymbol{\Phi}}, \hat{\boldsymbol{y}})$ as the criterion to select parameters, which is indirectly related to our evaluation criterion (i.e., accuracy). Specifically, we test multiple values of $c$ and $\ell$, and select the value that produces a model with the maximal likelihood on the given dataset. This method ensures us to select model without any prior knowledge on the true labels. For the special case of M³V, we fix the learned true labels $\boldsymbol{y}$ after training the model with certain parameters, and learn confusion matrices that optimize the full likelihood in Eq. (2).

Note that the likelihood-based cross-validation strategy [25] is not suitable for CrowdSVM, because this strategy uses marginal likelihood $p(\boldsymbol{X}|\boldsymbol{\Phi})$ to select model and ignores the label information of $\boldsymbol{y}$, through which the effect of constraints is passed for CrowdSVM. If we use this strategy on CrowdSVM, it will tend to optimize the generative component without considering the discriminant constraints, thus resulting in $c \to 0$, which is a trivial solution for model selection.

## 6.3 Experimental Results

We first test our estimators on the task of estimating true labels. For CrowdSVM, we set $\boldsymbol{\alpha} = \mathbf{1}$ and $v = 1$ for all experiments, since we find that the results are insensitive to them. For M³V, CrowdSVM and Gibbs-CrowdSVM, the regularization parameters $(c, \ell)$ are selected from $c = 2\hat{} [-8 : 0]$ and $\ell = [1, 3, 5]$ by the method in Sec. 6.2. As for Gibbs-CrowdSVM, we generate 50 samples in each run and discard the first 10 samples as burn-in steps, which are sufficiently large to reach convergence of the likelihood. The reported error rate is the average over 5 runs.

Table 2 presents the error rates of various estimators. We group the comparisons into three parts:

I. All the MV, IWMV and M³V are purely discriminative estimators. We can see that our M³V produces consistently lower error rates on all the four datasets compared with the vanilla MV and IWMV, which show the effectiveness of max-margin principle for crowdsourcing;

II. This part analyzes the effects of prior and max-margin regularization on improving the DS model. We can see that DS+Prior is better than the vanilla DS model on the two larger datasets by using a Dirichlet prior. Furthermore, CrowdSVM consistently improves the performance of DS+Prior by considering the max-margin constraints, again demonstrating the effectiveness of max-margin learning;

III. This part compares our Gibbs-CrowdSVM estimator to the state-of-the-art minimax entropy estimators. We can see that Gibbs-CrowdSVM performs better than CrowdSVM on Web-Search, Age and Flowers datasets, while worse on the small Bluebuirds dataset. And it is comparable to the minimax entropy estimators, sometimes better with faster running speed as shown in Fig. 2 and explained below. Note that we only test Entropy (O) on two ordinal datasets, since this method is specifically designed for ordinal labels, while not always effective.

Fig. 2 summarizes the training time and error rates after each iteration for all estimators on the largest Web-Search dataset. It shows that the discriminative methods (e.g., IWMV and M³V) run fast but converge to high error rates. Compared to the minimax entropy estimator, CrowdSVM is

Table 2: Error-rates (%) of different estimators on four datasets.

| | METHODS | WEB SEARCH | AGE | BLUEBIRDS | FLOWERS |
|---|---|---|---|---|---|
| I | MV | 26.90 | 34.88 | 24.07 | 22.00 |
| | IWMV | 15.04 | 34.53 | 27.78 | 19.00 |
| | M$^3$V | **12.74** | **33.33** | **20.37** | **13.50** |
| II | DS | 16.92 | 39.62 | 10.19 | **13.00** |
| | DS+PRIOR | 13.26 | 34.53 | 10.19 | 13.50 |
| | CROWDSVM | **9.42** | **33.33** | 10.19 | 13.50 |
| III | ENTROPY (M) | 11.10 | **31.14** | **8.33** | 13.00 |
| | ENTROPY (O) | 10.40 | 37.32 | – | – |
| | G-CROWDSVM | **7.99 ± 0.26** | 32.98 ± 0.36 | 10.37±0.41 | **12.10 ± 1.07** |

computationally more efficient and also converges to a lower error rate. Gibbs-CrowdSVM runs slower than CrowdSVM since it needs to compute the inversion of matrices. The performance of the DS estimator seems mediocre — its estimation error rate is large and slowly increases when it runs longer. Perhaps this is partly because the DS estimator cannot make good use of the initial knowledge provided by majority voting.

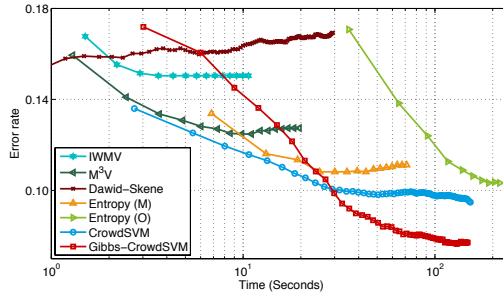

Figure 2: Error rates per iteration of various estimators on the web search dataset.

We further investigate the effectiveness of the generative component and the discriminative component of CrowdSVM again on the largest Web-Search dataset. For the generative part, we compared CrowdSVM ($c = 0.125, \ell = 3$) with DS and M$^3$V ($c = 0.125, \ell = 3$). Fig. 3(a) compares the negative log likelihoods (NLL) of these models, computed with Eq. (2). For M$^3$V, we fix its estimated true labels and find the confusion matrices to optimize the likelihood. The results show that CrowdSVM achieves a lower NLL than DS; this suggests that by incorporating M$^3$V constraints, CrowdSVM finds a better solution of the true labels as well as the confu-

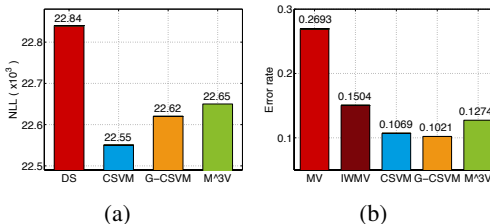

Figure 3: NLLs and ERs when separately test the generative and discriminative components.

sion matrices than that found by the original EM algorithm. For the discriminative part, we use the mean of worker weights $\hat{\boldsymbol{\mu}}$ to estimate the true labels as $y_i = \mathrm{argmax}_{d \in [D]} \hat{\boldsymbol{\mu}}^\top \boldsymbol{g}(\boldsymbol{x}_i, d)$, and show the error rates in Fig. 3(b). Apparently, the weights learned by CrowdSVM are also better than those learned by the other MV estimators. Overall, these results suggest that CrowdSVM can achieve a good balance between the generative modeling and the discriminative prediction.

## 7 Conclusions and Future Work

We present a simple and intuitive max-margin majority voting estimator for learning-from-crowds as well as its Bayesian extension that conjoins the generative modeling and discriminative prediction. By formulating as a regularized Bayesian inference problem, our methods naturally cover the classical Dawid-Skene estimator. Empirical results demonstrate the effectiveness of our methods.

Our model is flexible to fit specific complicated application scenarios [22]. One seminal feature of Bayesian methods is their sequential updating. We can extend our Bayesian estimators to the online setting where the crowdsourcing labels are collected in a stream and more tasks are distributed. We have some preliminary results as shown in Appendix B. It would also be interesting to investigate more on active learning, such as selecting reliable workers to reduce costs [9].

**Acknowledgments**

The work was supported by the National Basic Research Program (973 Program) of China (Nos. 2013CB329403, 2012CB316301), National NSF of China (Nos. 61322308, 61332007), Tsinghua National Laboratory for Information Science and Technology Big Data Initiative, and Tsinghua Initiative Scientific Research Program (Nos. 20121088071, 20141080934).

## Footnotes

[1]A maximum entropy estimator can be understood as a dual of the MLE of a probabilistic model [6].

[2]The offset $b$ is canceled out in the margin constraints.

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
