[Supplementary Material · mmmv15_supp.pdf]

# Max-Margin Majority Voting for Learning from Crowds Supplementary Materials

## A Posterior Inference for Gibbs-CrowdSVM Estimator

**Sample $\boldsymbol{\Phi}$:** The posterior of $\boldsymbol{\Phi}$ is similar as in CrowdSVM. We can derive that $q(\boldsymbol{\phi}_{mk}|\boldsymbol{y}) = \mathrm{Dir}(\boldsymbol{\phi}_{mk}|\boldsymbol{n}_{mk} + \boldsymbol{\alpha})$, which is also a Dirichlet distribution.

**Sample $\boldsymbol{\eta}$:** Given its prior $p_0(\boldsymbol{\eta}) = \mathcal{N}(\boldsymbol{\eta}; \mathbf{0}, v\boldsymbol{I})$, we can show that the posterior is also a normal distribution, $q(\boldsymbol{\eta}|\boldsymbol{y}, \boldsymbol{\lambda}) = \mathcal{N}(\boldsymbol{\eta}; \boldsymbol{\mu}, \Sigma)$, whose mean is $\boldsymbol{\mu} = \Sigma \left( c^2 \sum_i (\frac{1}{c} + \ell \lambda_i^{-1}) \boldsymbol{g}_i^{\Delta}(s_i) \right)$ and covariance matrix is $\Sigma = \left( \frac{1}{v}\boldsymbol{I} + c^2 \sum_i \frac{1}{\lambda_i} \boldsymbol{g}_i^{\Delta}(s_i) \boldsymbol{g}_i^{\Delta}(s_i)^{\top} \right)^{-1}$.

**Sample $\boldsymbol{\lambda}$:** The conditional distribution of the augmented variables $\boldsymbol{\lambda}$ is $q(\boldsymbol{\lambda}|\boldsymbol{\eta}, \boldsymbol{y}) = \prod_i q(\lambda_i|\boldsymbol{\eta}, y_i)$, where $q(\lambda_i|\boldsymbol{\eta}, y_i) = \mathcal{GIG}(\lambda_i; -\frac{1}{2}, 1, c^2(\ell - \boldsymbol{\eta}^{\top}\boldsymbol{g}_i^{\Delta}(s_i))^2)$ is a generalized inverse Gaussian distribution. Thus, its inverse value follows an inverse Gaussian distribution

$$p(\lambda_i^{-1}|\boldsymbol{\eta}, y_i) = \mathcal{IG}\left( \lambda_i^{-1}; |c(\ell - \boldsymbol{\eta}^{\top}\boldsymbol{g}_i^{\Delta}(s_i))|^{-1}, 1 \right), \tag{16}$$

from which we can draw samples efficiently [13], with $\mathcal{O}(1)$ time complexity.

**Sample $\boldsymbol{y}$:** Finally, for the true labels we can derive that each single variable $y_i$ follows a multinomial distribution:

$$q(y_i = d|\boldsymbol{\Phi}, \boldsymbol{\eta}, \lambda_i) \propto \widetilde{p}(\boldsymbol{x}_i, \lambda_i|\boldsymbol{\Phi}, \boldsymbol{\eta}, y_i = d). \tag{17}$$

With the conditional distributions, we iterate the above steps for a number of rounds to take samples from the posterior. In our experiments, we initially set $\boldsymbol{y}$ as the result of majority voting. After removing burn-in samples, we use the most frequent value of $\boldsymbol{y}$ as the final outputs.

## B Online Learning from Crowds

We further extend our methods to an interesting new setting of online learning-from-crowds, where the crowdsourcing labels are collected in a stream as more tasks are distributed over the crowdsourcing platforms. This setting is useful for many applications [1] that require a timely response, for which we cannot simply wait for collecting all the data before learning a model. We demonstrate that our online models can immediately estimate the true labels of current tasks, without caching all historical worker labels.

One seminal feature of Bayesian methods is their sequential updating. We now briefly discuss an extension of our Bayesian max-margin estimator to the interesting setting of online learning-from-crowds, where workers continue labeling the tasks that come in a stream. This setting is of interest in the case where the data are generated in a stream and our model needs to respond rapidly to provide reliable answers for further decisions. For example, a never-ending-learning system [1] may continue generating some suspicious facts that call for human labels in order to justify their confidence. In this case, an online learning-from-crowds model would be useful to quickly collect the crowd labels and estimate the true answers. Another scenario is the incremental learning for real-time applications, such as fraud protection, target marketing and intrusion detection, in which the underlying data distribution is likely to change. They require to update models with the sequentially arriving data to keep the prediction results accurate. Thus collecting new labels quickly and accurately is important to them.

Below, we present the online CrowdSVM estimator and the online Gibbs-CrowdSVM estimator, which are the extensions for CrowdSVM and Gibbs-CrowdSVM.

### B.1 Online CrowdSVM Estimator

We consider the online setting where a mini-batch of tasks, $\mathcal{B}_t$, are distributed to the workers for labeling, and the goal is to estimate the true labels of these tasks from the observations $\boldsymbol{X}_t$. We

assume that the set of workers doesn't change[3]. Let $q_{t-1}(\mathbf{\Phi}, \boldsymbol{\eta})$ be the inferred posterior after seeing the $(t\text{-}1)$th mini-batch. Then, given the new mini-batch of tasks $\mathcal{B}_t$, we define the online CrowdSVM as solving:

$$\inf_{\xi_i \geq 0, q_t, \boldsymbol{y}_t} \mathcal{L}^t \left( q_t(\mathbf{\Phi}, \boldsymbol{\eta}); \boldsymbol{y}_t \right) + c \cdot \sum_{i \in \mathcal{B}_t} \xi_i \tag{18}$$

$$\text{s. t.}: \mathbb{E}_{q_t}[\boldsymbol{\eta}^\top \boldsymbol{g}_i^\Delta(d)] \geq \ell_i^\Delta(d) - \xi_i, \forall i \in \mathcal{B}_t, d \in [D],$$

where $\mathcal{L}^t(q_t; \boldsymbol{y}_t) = \text{KL}\left[q_t \| q_{t-1}\right] - \mathbb{E}_{q_t}\left[\log p(\boldsymbol{X}_t | \mathbf{\Phi}, \boldsymbol{y}_t)\right]$. If we ignore the crowdsourcing margin constraints (e.g., setting $c = 0$), solving the problem gives exactly the same posterior as doing the standard sequential Bayesian updating. By considering the extra margin constraints, our work represents an application of the streaming RegBayes [16] theory to online learning-from-crowds.

**Variational Inference.** To solve the problem, we introduce the mean-field assumption $q_t(\mathbf{\Phi}, \boldsymbol{\eta}) = q_t(\mathbf{\Phi})q_t(\boldsymbol{\eta})$, and develop an iterative procedure that alternatively updates each factor distribution:

**Update $q_t(\mathbf{\Phi})$:** Fixing the distribution $q_t(\boldsymbol{\eta})$ and true labels $\boldsymbol{y}$, the optimal solution has the closed-form: $q_t^*(\mathbf{\Phi}) \propto q_{t-1}(\mathbf{\Phi})p(\boldsymbol{X}_t | \mathbf{\Phi}, \boldsymbol{y}_t)$. Since the prior $p_0(\mathbf{\Phi})$ is a Dirichlet distribution, by induction the inferred distribution at each round is also Dirichlet: $q_t^*(\phi_{jk}) = \text{Dir}(\boldsymbol{n}_{jk}^t + \boldsymbol{\alpha})$, where $n_{jkd}^t = n_{jkd}^{(t-1)} + \sum_{i \in \mathcal{B}_t} \mathbb{I}(y_i = k, x_{ij} = d)$ with the initial condition that $n_{jkd}^0 = 0$.

**Update $q_t(\boldsymbol{\eta})$:** Given the distribution $q_t(\mathbf{\Phi})$ and true labels $\boldsymbol{y}$, this substep involves solving:

$$\inf_{\xi_i \geq 0, q_t(\boldsymbol{\eta})} \text{KL}\left[q_t(\boldsymbol{\eta}) \| q_{t-1}(\boldsymbol{\eta})\right] + c \cdot \sum_{i \in \mathcal{B}_t} \xi_i \tag{19}$$

$$\text{s. t.}: \mathbb{E}_{q_t}[\boldsymbol{\eta}^\top \boldsymbol{g}_i^\Delta(d)] \geq \ell_i^\Delta(d) - \xi_i, \forall i \in \mathcal{B}_t, d \in [D].$$

Let $\boldsymbol{\omega}$ be the Lagrange multipliers. The solution is $q_t^*(\boldsymbol{\eta}) \propto q_{t-1}(\boldsymbol{\eta}) \exp\left(\sum_{i \in \mathcal{B}_t} \sum_d \omega_i^d \boldsymbol{\eta}^\top \boldsymbol{g}_i^\Delta(d)\right)$. When the prior $p_0(\boldsymbol{\eta})$ is normal, by induction this posterior is also a normal distribution $q_t^*(\boldsymbol{\eta}) = \mathcal{N}(\boldsymbol{\mu}_t, v\boldsymbol{I})$, where the mean is sequentially updated as $\boldsymbol{\mu}_t = \boldsymbol{\mu}_{t-1} + \sum_{i \in \mathcal{B}_t} \sum_d \omega_i^d \boldsymbol{g}_i^\Delta(d)$ with the initial condition $\boldsymbol{\mu}_0 = \mathbf{0}$. Plugging the normal distributions $q_t$ and $q_{t-1}$ into problem (19) and absorbing the slacking variables, we can show that the optimal $\boldsymbol{\mu}_t$ is the solution of the following convex problem:

$$\inf_{\boldsymbol{\mu}} \frac{\|\boldsymbol{\mu} - \boldsymbol{\mu}_{t-1}\|^2}{2v} + c \sum_{i \in \mathcal{B}_t} \sum_d \left(\ell_i^\Delta(d) - \boldsymbol{\mu}^\top \boldsymbol{g}_i^\Delta(d)\right)_+,$$

which can be solved by a (sub)-gradient descent method.

**Update labels $\boldsymbol{y}_t$:** Given the distribution $q_t(\mathbf{\Phi}, \boldsymbol{\eta})$, the problem of updating labels is exactly the same as in the batch CrowdSVM. So we can use the same discriminant function to find the best label estimation of each task in the current mini-batch.

In summary, when a new mini-batch of tasks comes, we iterate the above steps until converge, and get the estimated true labels at the final iteration.

## B.2 Online Gibbs-CrowdSVM Estimator

Similar as the Gibbs-CrowdSVM estimator, we use the expected loss for posterior regularization to replace the average loss of the online CrowdSVM estimator. When given the new mini-batch of tasks $\mathcal{B}_t$, this replacement leads to online Gibbs-CrowdSVM problem:

$$\inf_{q \in \mathcal{P}} \mathcal{L}^t(q_t(\mathbf{\Phi}, \boldsymbol{\eta}, \boldsymbol{y}_t)) + \mathbb{E}_q\left[\sum_{i \in \mathcal{B}_t} 2c(\zeta_{is_i})_+\right], \tag{20}$$

where $\mathcal{L}^t(q_t) = \text{KL}\left[q_t \| q_{t-1}\right] - \mathbb{E}_{q_t}\left[\log p(\boldsymbol{X}_t | \mathbf{\Phi}, \boldsymbol{y}_t)\right]$, and $\zeta_{id} = \ell_i^\Delta(d) - \boldsymbol{\eta}^\top \boldsymbol{g}_i^\Delta(d)$, $s_i = \arg\max_{d \neq y_i} \zeta_{id}$. Note here we treat the true labels $\boldsymbol{y}_t$ as variables rather than parameters.

To solve the online Gibbs-CrowdSVM problem, we introduce the mean-field assumption that $q_t(\mathbf{\Phi}, \boldsymbol{\eta}, \boldsymbol{y}_t) = q_t(\mathbf{\Phi}, \boldsymbol{\eta})q_0(\boldsymbol{y}_t)$. And we include the augmented variables $\boldsymbol{\lambda}_t$ to explain the constraints. The optimal post-data posterior with augmented variables after processing $t$ mini-batches is derived as

$$q_t(\mathbf{\Phi}, \boldsymbol{\eta}, \boldsymbol{y}_t, \boldsymbol{\lambda}_t) \propto q_0(\boldsymbol{y}_t)q_{t-1}(\mathbf{\Phi}, \boldsymbol{\eta})p(\boldsymbol{X}_t|\mathbf{\Phi}, \boldsymbol{y}_t) \prod_{i \in \mathcal{B}_t} \psi(y_i, \lambda_i|\boldsymbol{x}_i, \boldsymbol{\eta}). \tag{21}$$

**Posterior inference.** When $t$-th mini-batch of tasks comes, we do Gibbs sampling to infer the post-data posterior distribution in Eq. (21). The main steps are detailed as follows.

**Sample global variables:** Fixing all other variables, the conditional distribution of $\mathbf{\Phi}$ is

$$q_t(\boldsymbol{\phi}_{mk}|\boldsymbol{y}_t) \propto q_{t-1}(\boldsymbol{\phi}_{mk})p(\boldsymbol{X}_t|\mathbf{\Phi}, \boldsymbol{y}_t). \tag{22}$$

Similar as in the batch Gibbs-CrowdSVM, we can derive that $q_t(\boldsymbol{\phi}_{mk}|\boldsymbol{y}_t) = \mathrm{Dir}(\boldsymbol{\phi}_{mk}|\boldsymbol{n}_{mk}^t + \boldsymbol{\alpha})$. Given $p_0(\boldsymbol{\eta}) = \mathcal{N}(\boldsymbol{\eta}; \boldsymbol{0}, v\boldsymbol{I})$, the conditional distribution for $\boldsymbol{\eta}$ is

$$q_t(\boldsymbol{\eta}|\boldsymbol{y}_t, \boldsymbol{\lambda}_t) \propto q_{t-1}(\boldsymbol{\eta}) \prod_{i \in \mathcal{B}_t} \psi(y_i, \lambda_i|\boldsymbol{x}_i, \boldsymbol{\eta}), \tag{23}$$

which can be further derived as $q_t(\boldsymbol{\eta}|\boldsymbol{y}_t, \boldsymbol{\lambda}_t) = \mathcal{N}(\boldsymbol{\eta}; \boldsymbol{\mu}_t, \Sigma_t)$. The distribution mean is $\boldsymbol{\mu}_t = \boldsymbol{B}_t^{-1}\boldsymbol{A}_t$ and the covariance matrix is $\Sigma_t = \boldsymbol{B}_t^{-1}$, where $\boldsymbol{A}_0 = \boldsymbol{0}$ and $\boldsymbol{B}_0 = \frac{1}{v}\boldsymbol{I}$. The updating rules for these two notations are

$$\boldsymbol{A}_t = \boldsymbol{A}_{t-1} + c^2 \sum_{i \in \mathcal{B}_t} (\frac{1}{c} + \ell\lambda_i^{-1})\boldsymbol{g}_i^{\Delta}(s_i), \tag{24}$$

$$\boldsymbol{B}_t = \boldsymbol{B}_{t-1} + c^2 \sum_{i \in \mathcal{B}_t} \lambda_i^{-1}\boldsymbol{g}_i^{\Delta}(s_i)\boldsymbol{g}_i^{\Delta}(s_i)^{\top}. \tag{25}$$

**Sample local variables:** Fix the global variables $\mathbf{\Phi}$ and $\boldsymbol{\eta}$, the sampling procedure for the local variables $\boldsymbol{\lambda}_t$ and $\boldsymbol{y}_t$ is the same as that of the batch Gibbs-CrowdSVM in Eq. (16) and (17).

### B.3 Results in Online Learning

We test the online CrowdSVM on the web search dataset, which was split into a number of mini-batches. The regularization parameters are selected by the same method used for batch CrowdSVM. Since the data doesn't have specific ordering, we shuffle the mini-batches for 10 times and report the average results. For baselines, we compare with the online DS estimator, which is in fact a special case of our online CrowdSVM by simply setting $c = 0$.

Figure 4: Overall error rates of online CrowdSVM estimator with different mini-batch sizes.

**CrowdSVM.** Fig. 4 summarizes the overall error rates of CrowdSVM with different mini-batch sizes. We can see that when the mini-batch size increases, the overall error rate decreases. This is reasonable since larger mini-batches contain more information on the interrelationship between workers and tasks. Furthermore, the performance of batch CrowdSVM provides a lower bound for the performance of online estimators.

We further investigate the effectiveness of online estimators during the learning process. After processing each mini-batch, we fix the distribution $q(\mathbf{\Phi}, \boldsymbol{\eta})$, and estimate the true labels of the full

Figure 5: Online performances of different learning methods with various mini-batch sizes.

dataset. Fig. 5 shows the estimation error rates, with various mini-batch sizes. We also train a batch CrowdSVM on all the passed data after processing each mini-batch, whose performance acts as a lower bound of online CrowdSVM's error rates. Firstly, we can see that the error rates of all estimators decrease when processed more data, this result shows that the online learners can truly pass information through the time. Secondly, the curve of online CrowdSVM is very close to the lower bound curve, suggesting the effectiveness of this estimator. Finally, the results again support our observation that the online estimator's performance will improve along with the mini-batch size increasing.

**Gibbs-CrowdSVM.** We investigate the effectiveness of online Gibbs-CrowdSVM on the web search dataset. Fig. 6(a) shows the overall error rates of online Gibbs-CrowdSVM with different mini-batch sizes. The results show that the error rate of online Gibbs-CrowdSVM decreases as the estimator processes more data.

Figure 6: (a) Performances of online Gibbs-CrowdSVM with various mini-batch sizes. (b) Overall error rates of online Gibbs-CrowdSVM with different mini-batch sizes.

Fig. 6(b) shows the performance of the online Gibbs-CrowdSVM with different mini-batch sizes. Different lines in this figure show the error rates of online estimators with different mini-batch sizes. Comparison with the performance of the online CrowdSVM in Fig. 5 and Fig. 4 shows that online Gibbs-CrowdSVM performs worse than the variational version when the mini-batch size is small, perhaps because the Gibbs sampler on these small mini-batches of tasks possesses more uncertainty (i.e., high variance) than the variational inference algorithm.

## Footnotes

[3]This is reasonable sine we can simply define the set containing all web workers. If some workers were not active, the algorithm will simply ignore them.