[Reviews · NeurIPS 2015]

Submitted by Assigned_Reviewer_1

This paper adopts the idea of "margin" from SVM and proposes two models: max-margin majority voting and CrowdSVM. The former is discriminative and the latter is generative. The authors provide a thorough discussion and comparison with DS model. However, discussion on variants of the DS model is limited.

For the related work, what is the relation between the proposed method with other variants of the DS estimator, such as [23][15][11][12]?

Does this framework satisfy some theoretical guarantee? Can DS theoretical analysis be applied? The improvement of the proposed methods seems not significant.

Presentation and typos: In 3.1, "Assume that both workers provide labels 3 and 1 to item i." This sentence is ambiguous. It means that worker 1 provides labels 3 and 1 to item i, and worker 2

provides labels 3 and 1 to item i. Based on the context, I think you want to say that workers provide labels 3 and 1 to item i respectively.
Summary: This paper is incremental on a well-established field. The idea of introducing "margin" into majority voting is interesting and the model is clever, but comparison with existing work is not thorough.

Submitted by Assigned_Reviewer_2

Very good paper.

The paper presents a max-margin algorithm for learning from annotations collected from crowdsourcing workers. The main challenge here is to design aggregation strategies to infer the unknown true labels from the noisy labels provided by ordinary subjects that are not experts to the task at hand.

The paper contains lots of technical substance: it presents a standard max-margin framework, generalizes it to a Bayesian formulation and presents both a sampling and a variational inference solutions to the latter. The experimental results are generally promising.
Summary: Very good paper. Lots of substance: important problem, max margin framework, Bayesian generalization, variotailnal and sampling based inference, god experimental results.

I would like to see it accepted.

Submitted by Assigned_Reviewer_3

This paper formulates learning from crowds as a max-margin learning problem. To further consider the uncertainty of the model, it provides two Bayesian solutions: a Bayesian mean estimator and a Gibbs estimator to the posterior regularization term induced from the hinge-loss term in the original max-margin learning problem. Overall, the paper demonstrates a new successful application within the "Regularized Bayesian" framework.

The discriminative function for each task and each label was defined and the resulting margin was characterized. The resulting CrowdSVM problem differs from multi-task SVM, in the aspect that multi-class SVM handles individual features, while CrowdSVM represent a single task as dependent points lies on the axises. To consider generative perspective, the regularized Bayesian framework was introduced. The inference techniques (both variational inference and Gibbs sampling based on data augmentation for SVMs) were borrowed from previous RegBayes papers.

I have some concerns in the experiments. Since the entropy-based methods are the most competitive ones with max-margin methods, it worth further comparing these two class of methods. The online learning results seem weird at the first sight. To my understanding, the workers were fixed and the tasks comes in a stream. Assuming the tasks are distributed in some regular patterns, then the online result should recover the batch result. However, in the experiments small batch sizes perform significantly worse. Why? The paper did not give a clear answer. Figure 5 in the supplementary material shows some fluctuation in performance, which can be smoothed by repeating experiments.

Although having some concern in the experiments, the paper provides a novel max-margin learning strategy for learning from crowd. It is another empirically successful application for Bayesian discriminative learning.
Summary: A new attempt of formulating learning from crowds as max-margin learning. The paper also provides several solutions that achieve the state-of-the-art performance on several real datasets.

Submitted by Assigned_Reviewer_4

This paper proposed a max-margin majority voting for crowdsourcing problems. A Bayesian generalization is further proposed, which can be inferred by an efficient Gibbs sampler. Experiments on some benchmarks validate the performance of the proposed methods.

First, it is not clear why the model in (4) can converge to a good solution. It is not well explained why we should minimise the norm of the weights. Also, it is not well motivated why the proposed method is better than the existing models that also learns different worker expertise.

Second, an important issue in learning from crowdsourced data is that the label noise in the crowdsourced data can be generated by many reasons. The proposed model can only model the different expertise of different workers, but not some other important reasons, such as different sample difficulties modelled in [19,20]. Also, in the experiment, such methods are not compared. Thus, it is not sure whether the proposed method truly outperforms the state-of-the-art.
Summary: Overall, this paper contains some interesting ideas. But its novelty should be better clarified and experimental design could be strengthened.

Author Feedback
Author rebuttal: We thank the reviewers for acknowledging our contributions and providing valuable comments. We'll further improve the paper in the final version. We address the detail comments below.

To R1:

Q1: Relation with variants of DS:
Our main goal is to provide a discriminative max-margin formulation, which is general and complementary to generative methods. For example, though we consider the vanilla DS in CrowdSVM for both clarity and space limit, other variants (e.g., [15,11]) can be naturally incorporated, as the RegBayes formulation (9) is generally applicable to any Bayesian models. Finally, the spectral initialization method [23] for confusion matrices can also be used to initialize the confusion matrices in CrowdSVM, so as the methods in [12]. We'll add the discussion in the final version.

Q2: Theoretical guarantee:
We agree that theoretical analysis is an important topic, which will be systematically investigated in our future work. Intuitively, [23] provides a finite-sample error bound for decomposable aggregation rule estimators under the DS model. It's likely that their results can be extended by taking the margin into consideration for our M^3V. For CrowdSVM that involves Bayesian max-margin learning, it is more challenging but PAC-Bayes theory can be possibly applied to bound the error rate.

Q3: Improvement not significant:
In fact, M^3V significantly outperforms other discriminative models on all tested datasets (See Table 2-I), which demonstrates the benefits of max-margin learning. And the comparisons with DS models and entropy-based methods show that CrowdSVM can achieve state-of-the-art accuracy (See Q1 of R3 for more discussion). We think (as agreed by other reviewers) these results are convincing.

Q4: Ambiguous sentence:
Thanks for pointing out. We'll correct it.

To R3:

Q1: Comparison with entropy-based methods:
We agree that entropy-based methods are most competitive. Since our main focus is to provide a new max-margin formulation, we included the current set of results (See Table 2) for both clarity and page limit. The results demonstrate that our max-margin methods are comparable to the state-of-the-art methods, sometimes better with faster running speed (See Fig. 2). We'd like to include more analysis in the final version.

Q2: Online learning results:
In our case, online learning doesn't reduce to its batch counterpart, except when the mini-batch is set to the full set. The slightly worse performance of online learning in Fig.5 is probably due to the inaccurate estimates: in batch-learning, all information is available to estimate parameters, while in online learning, only partial information is available in a mini-batch to update the parameters at each iteration. The estimation error can be accumulated, leading to slightly worse error rate. Similarly, if the mini-batch size is too small, the error generated in each epoch is larger, so the overall performance will be even worse.

Finally, we'll repeat more times to smoothen Fig. 5.

To R6 (light): differences across datasets:

Thanks for the suggestions. We'll add more analysis. One possible reason why generative methods are better on Bluebirds is that every worker provides label to each picture. This label redundancy helps estimate the workers' confusion matrix for generative models. The other datasets don't have such redundancy.

To R7 (light): justification of margin:

Thanks for the suggestion. In fact, our methods for real data use soft-margin (See Eq.s 5, 9 & 14), which surely exists. Another nice property of max-margin methods is the sparsity of support vectors (SV). We observed that about 7% and 4% points are SV on the two relatively large datasets, Web and Age, respectively. In addition, our results clearly show that Majority-Voting can truly benefit from max-margin learning.

To R8:

Q1: minimize the norm of the weights:
It's well-known in SVMs that minimizing the norm of the weights is equivalent to maximizing the margin of the decision boundary. We adopted the same principle to define Eq.4.

Q2: differences with other weighted methods:
M^3V is the first to introduce max-margin principle, which (arguably) has better discriminative ability to learn weights. Moreover, the M^3V estimator can be integrated with generative models to describe more flexible situations.

Q3: other possible factors:
We agree that considering more factors is possible to enhance the model; but a complex model also has a high risk of overfitting. In fact, many complex models with lots of factors have been shown to be worse than simple ones like MV and DS (See [*]). To demonstrate the benefits of max-margin learning, we focused on the vanilla DS model, which has mild complexity. We'll examine more factors, which can be naturally incorporated into our general formulation (See Q1 to R1).

[*] Sheshadri, A., et al. Square: A benchmark for research on computing crowd consensus. 2013.